# Experimental Yellow Fever in Squirrel Monkey: Characterization of Liver In Situ Immune Response

**DOI:** 10.3390/v15020551

**Published:** 2023-02-16

**Authors:** Milene S. Ferreira, Jorge R. Sousa, Pedro S. Bezerra Júnior, Valíria D. Cerqueira, Carlos A. Oliveira Júnior, Gabriela R. C. Rivero, Paulo H. G. Castro, Gilmara A. Silva, José Augusto P. C. Muniz, Eliana V. P. da Silva, Samir M. M. Casseb, Carla Pagliari, Lívia C. Martins, Robert B. Tesh, Juarez A. S. Quaresma, Pedro F. C. Vasconcelos

**Affiliations:** 1Evandro Chagas Institute, Rodovia BR 316, km-07, Ananindeua 67030-000, Pará, Brazil; 2Postgraduate Program in Biology of Infectious and Parasitic Agents, Federal University of Pará, Belém 66075-110, Pará, Brazil; 3Laboratory of Animal Pathology, Institute of Veterinary Medicine, Federal University of Pará, Castanhal 68746-360, Pará, Brazil; 4National Primate Center, Ananindeua 67030-000, Pará, Brazil; 5Faculty of Medicine, University of Sao Paulo, Sao Paulo 01246-903, SP, Brazil; 6Department of Pathology, Center for Tropical Diseases, University of Texas Medical Branch, Galveston, TX 77555-0419, USA; 7Tropical Medicine Center, Federal University of Pará, Belém 66055-240, Pará, Brazil; 8Department of Pathology, Pará State University, Belém 66050-540, Pará, Brazil

**Keywords:** yellow fever, pathogenesis, experimental infection, immunology of yellow fever virus infection, yellow fever virus, *Saimiri* spp.

## Abstract

Non-human primates contribute to the spread of yellow fever virus (YFV) and the establishment of transmission cycles in endemic areas, such as Brazil. This study aims to investigate virological, histopathological and immunohistochemical findings in livers of squirrel monkeys (*Saimiri* spp.) infected with the YFV. Viremia occurred 1–30 days post infection (dpi) and the virus showed a predilection for the middle zone (Z2). The livers were jaundiced with subcapsular and hemorrhagic multifocal petechiae. Apoptosis, lytic and coagulative necrosis, steatosis and cellular edema were also observed. The immune response was characterized by the expression of S100, CD11b, CD57, CD4 and CD20; endothelial markers; stress and cell death; pro and anti-inflammatory cytokines, as well as Treg (IL-35) and IL-17 throughout the experimental period. Lesions during the severe phase of the disease were associated with excessive production of apoptotic pro-inflammatory cytokines, such as IFN-γ and TNF-α, released by inflammatory response cells (CD4+ and CD8+ T lymphocytes) and associated with high expression of molecules of adhesion in the inflammatory foci observed in Z2. Immunostaining of the local endothelium in vascular cells and the bile duct was intense, suggesting a fundamental role in liver damage and in the pathogenesis of the disease.

## 1. Introduction

Yellow fever remains a significant public health problem in tropical region of Africa and South America. The disease is endemic in 45 countries (11 South American; 34 African), which periodically experience outbreaks and where, according to the World Health Organization, it is consistently underreported [1,2].

The yellow fever virus (YFV) infection in humans can be asymptomatic or symptomatic. Approximately 60–80% of infections are asymptomatic or oligosymptomatic. Clinically, it can be mild, moderate, or severe with a high fatality rate (50%). Most severe cases present with the classical signs of hepatic and renal failure, including jaundice, oliguria/anuria, albuminuria, and hemorrhages. The severe form of the disease is characterized by simultaneous involvement of several organs [3,4].

Understanding the YFV–host relationship is essential to comprehension of the mechanisms that affect the innate and adaptive immune responses to the virus infection. Studies with humans and experimental animal models have shown that the liver is the main target organ of YFV. Hepatic lesions appear to be caused by both the direct cytopathic effects and the immune response induced by cytokines and immune cells. These effects have been well characterized in previous studies [5,6], that demonstrated the histopathological features found in human liver samples were related not only to the presence of viral antigen in the hepatic tissue, but also to the expression of several cytokines, mainly TNF-α and TGF-β, and their relationships with the apoptotic cellular immune response are particularly important in the development of hepatic lesions.

The epidemiological relevance of yellow fever and its importance for public health are justified by the epidemic potential of the YFV, mainly in urban areas, where the high rate of *Aedes aegypti* and the high population contingent of unvaccinated susceptible are of concern [7]. The use of experimental animal models of the YFV infection in neotropical non-human primates (NHPs), may help to define the roles of cytokines and other factors that are involved in responses to different clinical forms of disease, thereby improving our understanding of the pathogenesis of the disease. In this study, we characterized the hepatic tissue changes and in situ immune responses in an experimental squirrel monkey (*Saimiri* spp.) model of YFV infection, using a South American genotype I strain of virus. Squirrel Monkey are widely distributed in the Amazon region in Northern South America, which increase their importance as primary hosts, viral amplifiers, and sources of information on the dynamics of the circulation of YFV during and outside of epidemics.

## 2. Materials and Methods

### 2.1. Samples

Eleven laboratory-reared squirrel monkeys (*Saimiri* spp.) from a breeding colony of the National Primate Center (CENP) Ananindeua, Pará, Brazil were used in the study. Before the infection all animals were bled and tested negative by hemagglutination-inhibition (HI) test for the presence of antibodies against YFV, dengue virus 1–4, Ilheus virus, Rocio virus, and Sant Louis virus. The non-immune animals were intradermally infected with YFV genotype I isolate BeH655417 (infectious dose: 1 × 10^6^ plaque forming units (PFU)/mL). One animal was not infected and served as a negative control. The YFV sample used in this study was originally obtained from a severe fatal human case in Roraima, Brazil in 2014 and was genetically characterized as of South American genotype I (BeH655417) and isolated in the Evandro Chagas Institute (IEC), Ananindeua, Pará, Brazil and propagated by a single additional passage in C6/36 cells. Infection by YFV cells was confirmed by indirect immunofluorescence assay with polyclonal and monoclonal anti-YFV antibodies [8].

One monkey was euthanized each day during the first 7 days and at 10-, 20-, and 30-days post-infection (dpi). For necropsy, the animals were euthanized intravenously with ketamine (15 mg/kg) and xylazine (1 mg/kg) [9]. Subsequently, macroscopic examination and photographic documentation (Fujifilm S290, Tokyo, Japan) was performed. Liver tissue was collected as three individual samples: two were preserved at −70 °C and the third was fixed in 10% buffered formalin for 24 h and then stored in 70% ethanol until processing.

### 2.2. Viral RNA Detection

RNA extraction was done with a Maxwell^®^ extraction platform (Promega, Madison, WI, USA) and the commercially available Maxwell^®^ 10 LEV simply RNA Tissue Kit according to the manufacturer’s protocol. For the quantification of viral load, was used a commercial GoTaq Probe 1-Step RT-qPCR System kit (Promega) in conjunction with the absolute quantification method using a pGEM^®^-T Easy vector-cloned plasmid (Promega) from the YFV genome as previously described [10].

### 2.3. Histological and Immunohistochemical Assays for Detection of Viral Antigen and Cytokines

For histopathological analysis, 5-μm-thick histological sections of paraffin-embedded tissue specimens were stained with hematoxylin–eosin (HE). Immunohistochemistry (IHC) was adapted according to the previously described protocol [11]. An anti-YFV antibody produced in Balb-C mice was prepared by the IEC and included in a streptavidin–alkaline phosphatase assay. For the other markers, we performed an IHC assay for detection, based on complex formation with EnVision horseradish peroxidase (HRP)–polymer (Agilent, Santa Clara, CA, USA). Briefly, tissue samples were dewaxed in xylol and rehydrated in ethyl alcohol (90%, 80%, and then 70%). Endogenous peroxidase was inhibited by incubating the tissue in 3% hydrogen peroxide for 45 min. Antigen retrieval was performed by heating the sample in citrate buffer (pH 6.0) for 20 min at 90 °C. Nonspecific protein binding was blocked by incubating in 10% concentrated skim milk for 30 min. The histological sections were incubated at room temperature with primary antibodies diluted in 1% bovine albumin for 14 h. The slides were immersed in phosphate-buffered saline and incubated with the secondary EnVision HRP–polymer-conjugated antibody in a 37 °C oven for 30 min. The sections were developed with a chromogen solution consisting of 0.03% diaminobenzidine and 3% hydrogen peroxide. Tissue staining was carried out using Harris hematoxylin for 1 min. Finally, the histological sections were mounted in slides and analyzed in light microscopes. Relation of primary antibodies used in immunohistochemical reactions are available in the Appendix A.

### 2.4. Quantitative Analysis

For quantitative analysis, we analyzed samples with the Axio Imager Z1 optical microscope (Carl Zeiss, Ober Kochen, Germany) using a 1 cm^2^ graduated reticle (area: 0.0625 mm^2^) under a 400× objective. Immunologically marked cells were counted in 10 hepatic acini subdivided into the periportal (Z1), midzone (Z2), and central vein (Z3) areas and the portal tract (PT). We calculated the mean number of cells in each area and divided the result by 0.0625 mm^2^ [12] (Appendix A).

### 2.5. Statistical Analysis

The data obtained in the experiments were stored on electronic spreadsheets. We performed statistical analysis with GraphPad Prism version 5.0 (GraphPad, San Diego, CA, USA). A descriptive statistic was applied where, in the univariate analysis, frequencies and measures of central tendency were obtained. Values of *p* < 0.05 were considered statistically significant.

## 3. Results

### 3.1. Viral Kinetics and Macroscopic, Histopathological, and Immunohistochemical Detection of YFV Infection

The macroscopic changes caused by YFV infection in the squirrel monkeys followed the classical pattern. The livers were of normal size. However, they were diffusely jaundiced (2–10 dpi) during the acute phase (1–7 dpi) and at the onset of the convalescence phase (10–30 dpi). We observed a heterogeneous pattern in the hepatic tissue with irregular red areas amid lighter areas (4–5 dpi) with multifocal distribution, as well as subcapsular multifocal petechiae (7 dpi), and severe organ bleeding (6 dpi). In the convalescence phase, these alterations had disappeared or were minimal (20–30 dpi). One infected animal died on 6 dpi (Figure 1A).

The YFV genome was only detected in the monkeys during the acute phase of infection. The peak viremias occurred at 4 dpi (803,000 copies/μL) and 5 dpi (304,000 copies/μL) (Figure 1B). Viral antigens were predominantly detected in Z2 in hepatocytes and Kupffer cells. YFV tropism for Z2 progressed during the acute phase, leading to increased immunostaining at 4 to 6 dpi (Figure 1C).

Histopathological analysis revealed portal spaces that generally presented with a mild or moderate inflammatory infiltrate of lymphocytes, plasma cells, and macrophages that were disproportionate to the degree of hepatic parenchyma impairment. Hepatocyte vacuolation, coagulative and lytic necrosis, and apoptosis were associated with periportal mononuclear infiltrates, congestion, and hemorrhages during the acute phase (1–7 dpi). (Figure 1D–G).

In areas with centrilobular and midzonal lesions, most hepatocytes were granulated with an atypical eosinophilic cytoplasm and showed signs of karyolysis, pycnosis, and karyorrhexis. Hypertrophy and hyperplasia of Kupffer cells and moderate endothelial swelling were observed and were most evident in Z2. In Z1, the remaining hepatocytes showed swelling and marked vacuolization of the cytoplasm. The main findings in the convalescence phase were hepatocyte swelling and the presence of portal inflammatory infiltrates, which decreased up to 30 dpi (Figure 1D–G).

### 3.2. Characterization of In Situ Immune Cell Phenotypes and Protein Expression Profiles in the Hepatic Parenchyma

We characterized the in situ immune cell phenotypes in YFV-infected squirrel monkey livers using commercial antibodies against S100 (APC), CD11b (Macrophages), CD57 (NK Cells), CD4 (T Cells), and CD20 (B Cells). The expression of all markers in infected animals was higher during the acute phase than in the control tissue and decreased during the convalescence phase, the exception was CD20, which was not detected at 6 dpi (Figure 2E,J). S100 was more highly expressed in the PT than in Z2, mainly in the bile duct (Figure 2A,F). In contrast, the expression of CD11b and CD57 was highest in Z2 (Figure 2B,G,C,H). CD4 markers of helper T cells were more highly expressed in Z2 (Figure 2D,I).

In addition, the endothelial activation and adhesion markers VCAM-1, ICAM-1, and VLA-4 were expressed not only by endothelial cells in Z1 and Z2, but also by Kupffer cells in the sinusoidal cords (Figure 3). We also evaluated the expression of the cell stress response and death markers Caspase 3, MLKL, iNOS and lysozyme after squirrel monkey experimental YFV infections. They were also most highly expressed in Z2, especially in hepatocytes and Kupffer cells during the acute phase of infection (Figure 4). Finally, the Th1 (IFN-γ, IFN-β, TNF-α, and IL-8), Th17 (IL-17), and Th2 (TGF-β, IL-10 and IL-4) and Treg (IL-35) cytokines were most highly expressed in the acute phase, mainly at 6 dpi, in areas with extensive tissue damage, such as Z2 (Figure 5 and Figure 6). They were also expressed in the convalescence phase, although to a lesser degree. An illustration of a negative control is included in Appendix A.

## 4. Discussion

The risk of YF reurbanization where *Aedes aegypti* is currently established, especially in endemic and populated developing countries such as Brazil, has generated worldwide concern. Despite the magnitude and rapid expansion of the recent yellow fever epidemic in Brazil, little is known about YFV dispersion between the primary hosts and their specific in situ immune responses [13]. We found that YFV infects squirrel monkeys with intense hepatic tropism in the first days of infection. From day 7 onward, which coincided with beginning of the convalescence phase, viral RNA decreased drastically, likely due the activation of the immune response. These results have been corroborated in previous experimental studies with other flaviviruses and older descriptions of hepatic experiments with YFV. In fact, squirrel monkeys infected with dengue virus (DENV) and Zika virus (ZIKV); rhesus macaques (*Macaca mullata*) infected with DENV, YFV, and ZIKV; or marmosets (*Callithrix penicillata)* infected with DENV and ZIKV all exhibit more pronounced viremia, biochemical, and tissue changes in the early days of infection than squirrel monkeys infected with YFV [14,15,16,17,18,19,20].

We found by IHC that YFV has a tropism for several resident cells of the hepatic parenchyma, with a clear preference for hepatocytes, Kupffer cells, bile duct cells, and vascular endothelial cells in the PT. Other studies have reported a similar YFV tropism for the liver, as corroborated by high viral antigen IHC staining in hepatocytes and Kupffer and endothelial cells, which likely serve as the primary sites for YFV replication in the liver excepting for bile duct cells that together with endothelial cells seems to have an important role in disease pathogenesis in hepatic tissues. In addition, Kupffer cells are a major population of antigen-presenting cells (APCs) in the hepatic environment; they present MHC II-linked viral antigens to CD4+ and CD8+ T lymphocytes in the hepatic inflammatory infiltrate during YFV infection [5,6,21,22].

The histopathological changes in the experimental model showed that the hepatic parenchyma lesions were most intense in Z2, as previously widely reported [1,5,21]. The animal that died at 6 dpi had major alterations in its hepatic tissue due to deleterious effects of the virus and showed that squirrel monkeys have a relative resistance to YFV infection with a small proportion of infected animals developing of severe fatal disease. On the other side, despite the decline of the viral load at 7 dpi, the hepatic lesions in the animals were pronounced and present variable intensities, which indicates the activation of the immune response, which should have a role in the resistance to YFV and helping of survival of many of infected animals.

As previously described in the human liver [1,5,21], the main type of YFV-induced cell death was apoptosis, followed by lytic and coagulative necrosis, as demonstrated by intense immunostaining for caspase 3 and MLKL, respectively. We observed irregular changes in caspase 3 staining over the course of the experiment. One hypothesis that accounts for this finding is that the vascular impairment associated with YFV results in intense tissue hypoxia, which may ultimately culminate in coagulative necrotic lesions. Indeed, intense midzone lesions have been associated with low-flow hypoxia. It is noteworthy, therefore, that the more pronounced midzone (Z2) lesions were also associated with YFV tropism for this zone, resulting in cell death by apoptosis and/or lytic necrosis, probably due to the cellular viral cycle and the host immune response locally mediated by cytokines such as TNF-α and TGF-β, as previously demonstrated in the YF liver of human fatal cases [1,5,21].

Necrosis and apoptosis are classically associated with infectious liver diseases [23,24]. Another mechanism of cell death, necroptosis, which has not previously been described during YF, appears to contribute to disease pathogenesis in our model, as characterized by intense immunostaining with an antibody against MLKL. Necroptosis or programmed necrosis is dependent on the formation of the necrosome, which includes RIP1, RIP3, and MLKL [25,26,27]. MLKL phosphorylation in the necrosome induces the activation of genes that cause cell death. Given that necrosis is a classical mechanism of YFV-induced cell death, it will be critical to investigate the mechanisms of necroptosis related to MLKL, which may modulate endosomal traffic and phagolysosome formation. We cannot rule out necroptosis-mediated TNF-α responses downstream of TRADD and iNOS activation and reactive oxygen species (ROS) generation [25,26,27], especially at 6 dpi when the liver parenchymal damage was the most intense.

We found immunostaining of innate and adaptive immune cells in the acini and PT through the acute-phase infection, particularly at 6 dpi. Interestingly, S100, a classical marker of APCs, was most highly expressed in the PT, predominantly in bile duct cells. We have also observed intense immunostaining of hepatic cells with the anti-YFV antibody in the PT. Previous study [28] with human tissues have reported the expression of S100 in the hepatic parenchyma, especially in the PT and Z2, marking APCs that along with Kupffer cells, play important roles in the processing and presentation of antigens to T lymphocytes during in situ immune responses after YFV infection. Although it has not been previously reported, the expression of S100 in the bile duct may indicate the involvement of endothelial cells in the duct, a local place for processing and presenting of YFV antigens. Classically, in certain conditions, endothelial cells may act as APCs, thereby enhancing the adaptive immune response by activating T lymphocytes [28].

We also investigated the role of Kupffer cells and perivascular macrophages in PT. CD11b labeling showed that these cells display an M1 phenotype in Z2 and the PT, as confirmed by their labeling for lysozyme and iNOS, enzymes responsible for generating nitric oxide and ROS that cause cellular damage. Kupffer cells are fundamental to the pathogenesis of YFV and other flavivirus infections, since the deleterious effects of YFV and the antiviral response are dependent on M1 macrophages and the cytokines that activate these cells, such as TNF-α, IFN-α, and IFN-γ [18,29,30].

The NK cell marker CD57+ was more highly expressed throughout the experiment in Z3 and, predominantly Z2 than in Z1, indicating the development of a pro-inflammatory innate antiviral response against the virus. Our findings are consistent with previous reports that YFV and other flavivirus-induced activation of NK cells increases the production of TNF-α and IFN-γ, thereby potentiating the Th1 response [31,32].

We observed CD4+ T cells with Th1, Th2, Th17, and regulatory T cell (Treg) phenotypes in the PT and hepatic acini. Notably, the expression of the cytokines that characterize the responses mediated by these cells was higher in Z2 than in the PT. Several studies have demonstrated that Th1 cytokines such as TNF-α, IFN-γ, and IFN-β contribute directly to trigger antiviral M1 macrophages and the development of a cellular stress environment [33,34,35]. The expression of these cytokines peaked at 6 dpi, when the lesions in the hepatic parenchyma were the most intense.

The cytokine TNF-α participates in the development of necroptosis and apoptosis; it is strongly associated with the regulation and activation of TRADD and the production of RIP1, RIP3, and caspase 3 [36,37]. In our study, NK cells appeared to be one of the main cell types involved in the inhibition of viral replication and development of the proinflammatory response through the production of type 1 IFN, TNF-α, and IFN-γ [5,6,12]. We also investigated the role of IL-8, which had an irregular expression pattern throughout the experiment; it was predominantly found in Z2. IL-8 may act as a chemotactic factor for the recruitment of immune cells. In fatal YF, IL-8 has the potential to improve the response of TNF-α and IL-6 [6]. It also acts as a regulatory factor for RANTES and promotes the infiltration of NK cells, T lymphocytes, and B cells [38,39,40].

The responses mediated by Th17 lymphocytes are often related to Th1 responses. During the experiment, IL-17 expression increased and peaked at 6 dpi mainly in Z2. There have been few in situ studies on the role of IL-17 in YF pathogenesis. However, analyses of infections by other flaviviruses have demonstrated that TNF-α and IL-6 intensify tissue damage and promote the recruitment of neutrophils to infectious foci in the hepatic parenchyma [41,42].

Importantly, the endothelium plays a central role in the immunopathogenesis of viral hemorrhagic fevers, including YF [43]. Beyond being a tissue that undergoes intense changes during YF infection, especially during hemorrhagic infections, the endothelium also constitutes the gateway through which immune cells migrate into the hepatic parenchyma to mount and trigger the antiviral response. Thus, to investigate the relationship between the migration of immune cells and the expression of adhesion molecules, we observed the expression of VCAM-1, ICAM-1, and VLA-4 during the study. They were predominantly expressed in Z2 and the PT. We propose that the activation of endothelial cells may be directly related to the adhesion and transmigration of leukocytes to the infectious focus. Previous studies on DENV hepatic tropism have also shown elevated expression of VCAM-1, ICAM-1, and VLA-4 in infected compared to healthy tissues [44,45]. Other studies have linked midzonal lesions to alterations in the hepatic vascular bed and low-flow hypoxia, which would imply the development of characteristic, more intense lesions in Z2. In the YFV study, the roles of direct viral cytopathic and cellular immune effects in the induction of this lesion pattern were demonstrated by high expression of the YFV antigen in Z2 and the induction of an injury of the immune system in this region [15,16,43].

Th2 and Treg responses antagonize Th1 and Th17 responses. We found that Th2 and Treg responses contribute to the regulation of the antiviral inflammatory response and the consequent preservation of the liver parenchyma. Indeed, the expression kinetics of IL-4, IL-10, TGF-β, and IL-35 were similar to those of the Th1 cytokines; they were predominantly expressed in Z2 and peaked at 6 dpi. Among these cytokines, IL-4 is classically incriminated as an inhibitor of Th1 responses and potentiates of the CD20+ B cell-mediated humoral responses that are fundamental to plasma cell activation and the production of protective anti-YFV neutralizing antibodies.

IL-10 and IL-35 have important regulatory roles for effective antiviral responses. These cytokines maintain the appropriate response intensity to clear the viral agent and prevent the development of an immune response that may lead to secondary tissue involvement, especially in hepatic and endothelial tissues, which could induce hemorrhage [46,47,48]. IL-35 was expressed in all zones with peak expression also in Z2 at 6 dpi. Studies of other flaviviruses have shown that Treg and Th2 responses synergize in the hepatic parenchyma to control the deleterious effects mediated by Th1 lymphocytes and M1 macrophages that are responsible for the antiviral immune response during infection [49,50,51]. In this study, we found that TGF-β seemed to inhibit the activity of M1 macrophages and Th1 lymphocytes and helps to promote an environment that favors cell death by apoptosis, especially the YFV infected cells. Previous studies of livers from humans that died of yellow fever [5,6] and dengue [52] showed that TGF-β acts as a potent inducer of apoptosis in the hepatic parenchyma. It is also responsible for the characteristic disproportionality between the intensity of the inflammatory infiltrate and the degree of hepatic impairment, as the cytokine is able to inhibit cellular immune responses and strongly induce apoptosis [5,52,53,54,55] (Figure 7).

The limited number of NHP did not allow comparative analyzes between infected animals and the NC. It was not possible to collect samples from NHP livers before infection. This made it impossible to compare the expression of the studied markers before and after infection. Due to these limitations, we suggest that the results be interpreted with caution. The use of a single NHP per day of kinetics was due to the difficulty in obtaining the NHP studied, as well as the recommendation of the rational use of animals by the Ethics Committees.

## 5. Conclusions

In conclusion, this is the first experimental study in recent years to show that neotropical squirrel monkeys (*Saimiri* spp.) are susceptible to YFV infection, during which they develop an exquisite immune response and liver lesions that resemble those associated with human YFV and DENV infections. In this context, the present model can reliably reproduce the previously described responses in the livers of humans with YF and, thus, constitutes an excellent model for controlled experimental immunologic studies on YFV. The limited number of NHP was a limitation of the study. The data must be interpreted with caution. However, its relevance must be considered as it reproduces findings reported in humans. In summary, the results obtained in this experimental study contribute considerable new information to further our better understanding of yellow fever pathogenesis.

## Figures and Tables

**Figure 1 viruses-15-00551-f001:**
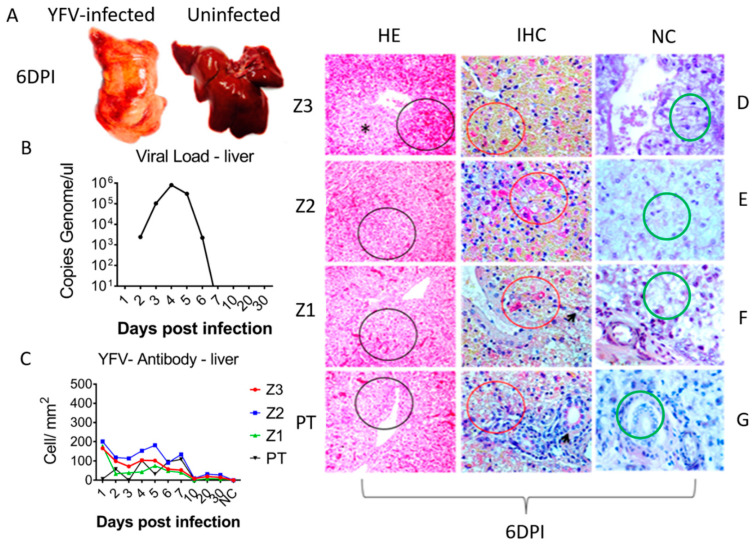
Gross macroscopy, viral load, histopathology and immunohistochemical assay in hepatic parenchyma of squirrel monkey (*Saimiri* spp.) infected with YFV South American genotype I. (**A**) Macroscopic representation of liver infected with YFV at 6 dpi showing an icteric and hemorrhagic pattern compared to a normal uninfected liver. (**B**) Evolution of viral kinetics highlighting viremic peak at 4 dpi. (**C**) Quantification of cell expressing specific YFV antigens by Immunohistochemical assay (IHC) in the 4 hepatic compartments analyzed demonstrating a wide predominance in Z2 during the acute phase of infection. (**D**) Representative histopathology photographs showing the involvement of Z3, with emphasis to lytic necrosis (asterisks), intense hemorrhage (black circle), positive immunohistochemistry for presence of specific YFV antigens in hepatocytes (red circle). (**E**) Histopathology showing the major involvement of Z2 with lytic necrosis area (black circle), positive immunohistochemistry showing large amounts of YFV antigens in hepatocytes (red circle). (**F**) Representative histopathology photographs of Z1, showing areas of periportal lytic necrosis and hemorrhages (black cycle), positive immunohistochemical for defection of specific YFV antigens and a periportal infiltrate is highlighted (red circle) and Kupffer cells (arrow). (**G**) Representative histopathology photographs of portal tract (PT) showing inflammatory infiltrate (black circle), positive immunohistochemistry for YFV antigens periportal infiltrate (red circle) and in the bile duct (black arrow); (**D**–**G**) Negative control for YFV antigens in Z3, Z2, Z1, and PT.

**Figure 2 viruses-15-00551-f002:**
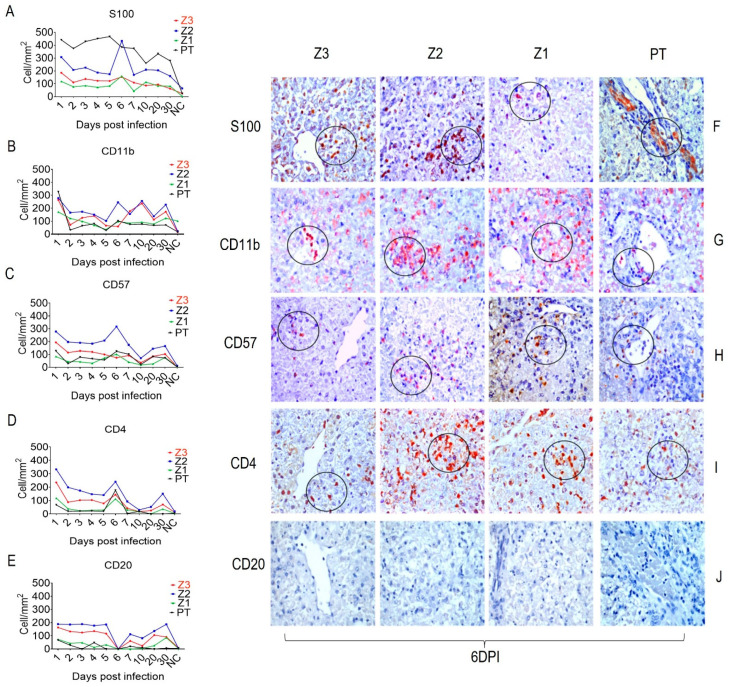
Quantitative immunohistochemical analysis for specific antibodies for immune cells in Z3, Z2, Z1 and PT cells in hepatic parenchyma of squirrel monkeys (*Saimiri* spp.) infected with YFV. Circles represent areas expressing higher immunostaining cells in different hepatic areas. (**A**) Expression of S100. (**F**) Immunostaining for S100 in Kupffer cells (F-Z3), inflammatory infiltrate (F-Z2), hepatocytes (F-Z1) and bile duct (F-PT). (**B**) Expression of CD11b. (**G**) Immunostaining for CD11b macrophages in inflammatory infiltrate (G-Z3), (G-Z2), (G-Z1), (G-PT). (**C**) Expression of CD57 NK cell. (**H**) Immunostaining for CD57 in inflammatory infiltrate in all hepatic compartments (H-Z3), (H-Z2), (H-Z1), (H-PT). (**D**) CD4 expression. (**I**) Immunostaining for CD4 T lymphocyte in the inflammatory infiltrate in all acini areas (I-Z3), (I-Z2), (I-Z1), (I-PT). (**E**) Expression of CD20 (B cell). (**J**) Absence of Immunostaining for CD20 in all acini areas (J-Z3), (J-Z2), (J-Z1), (J-PT). (**E**–**J**) obtained after 6 dpi.

**Figure 3 viruses-15-00551-f003:**
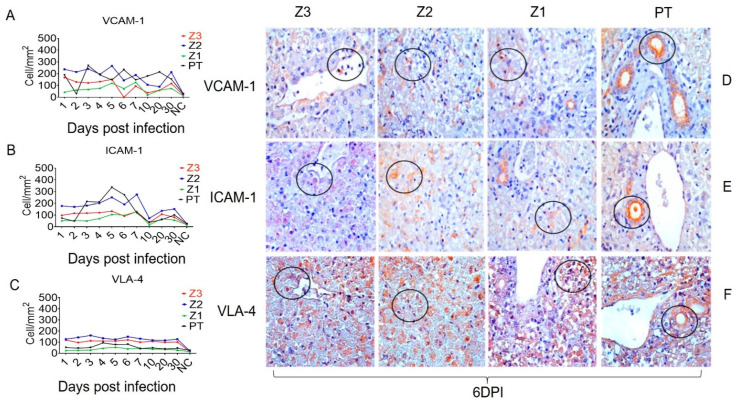
Quantitative immunohistochemical analysis of selected endothelial cell markers in zones Z3, Z2, Z1 and PT in hepatic parenchyma of squirrel monkeys (*Saimiri* spp.) infected with YFV. Circles represent areas expressing higher immunostaining cells in different hepatic areas. (**A**) Expression of VCAM-1. (**D**) Immunostaining for VCAM-1 in the inflammatory infiltrate (D-Z3), in Kupffer cells (D-Z2), (D-Z1) and bile duct (D-PT). (**B**) Expression of ICAM-1. (**E**) Immunostaining for ICAM-1 in hepatocytes (E-Z3), (E-Z2), (E-Z1) in Kupffer cells, (E-PT) in the bile duct. (**C**) VLA-4 expression. (**F**) Immunostaining for VLA-4 in endothelial and Kupffer (F-Z3) cells, (F-Z2) in inflammatory infiltrate, (F-Z1), (F-PT) in 6 dpi bile duct.

**Figure 4 viruses-15-00551-f004:**
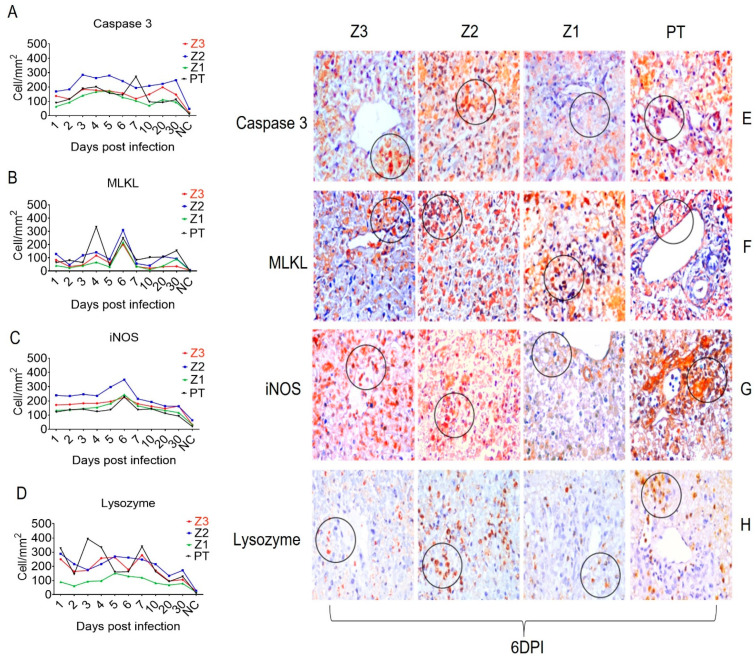
Quantitative immunohistochemical analysis of specific markers in Z3, Z2, Z1 and PT in the hepatic parenchyma of squirrel monkeys (*Saimiri* spp.) infected with YFV. Circles represent areas expressing higher immunostaining cells in different hepatic areas. (**A**) Caspase 3 expression. (**E**) Immunostaining for Caspase 3 in hepatocytes (E-Z3), (E-Z2), (E-Z1) and bile duct (E-PT). (**B**) MLKL expression. (**F**) Immunostaining for MLKL in hepatocytes and inflammatory infiltrate (F-Z3), (F-Z2) inflammatory infiltrate, (F-Z1) in hepatocytes, (F-PT) in hepatocytes. (**C**) Expression of iNOS. (**G**) Immunostaining for iNOS in inflammatory infiltrate (G-Z3), (G-Z2) in hepatocytes, (G-Z1) hepatocytes, (G-PT) bile duct and inflammatory infiltrate. (**D**) Lysozyme Expression. (**H**) In inflammatory infiltrate (E-Z3), (E-Z2), (H-Z1) in Kupffer cells, (H-PT) in inflammatory infiltrate at 6 dpi.

**Figure 5 viruses-15-00551-f005:**
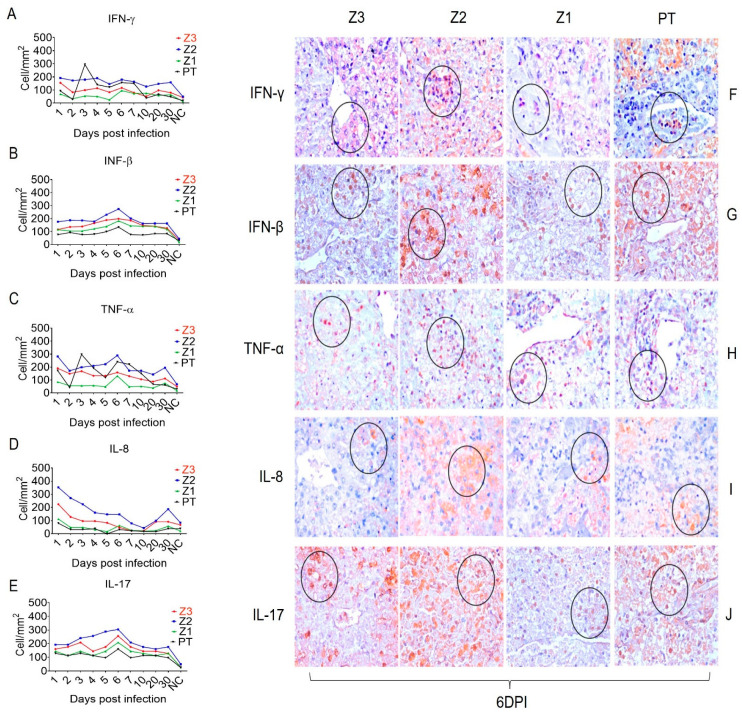
Quantitative immunohistochemical analysis of selected pro- inflammatory cytokines in zones Z3, Z2, Z1 and PT in the hepatic parenchyma of squirrel monkeys (*Saimiri* spp.) infected with YFV. Circles represent areas expressing higher immunostaining cells in different hepatic areas. (**A**) IFN-γ expression. (**F**)-Immunostaining for IFN-γ in hepatocytes (F-Z3), and in inflammatory infiltrate (F-Z2), (F-Z1), (F-PT). (**B**) Expression of IFN-β. (**G**) Immunostaining for IFN-β in hepatocytes (F-Z3), (F-Z2) in inflammatory infiltrate, (F-Z1) in hepatocytes, (F-PT). (**C**) Expression of TNF-α. (**H**) Immunostaining for TNF-α in inflammatory infiltrate (F-Z3), (F-Z2), (F-Z1), (F-PT). (**D**) Expression of IL-8. (**I**) Immunostaining for IL-8 in hepatocytes (F-Z3), (F-Z2), (F-Z1), (F-PT). (**E**) Expression of IL-17. (**J**) Immunostaining for IL-17 in hepatocytes (J-Z3), (J-Z2), (J-Z1), (J-PT) in 6 dpi.

**Figure 6 viruses-15-00551-f006:**
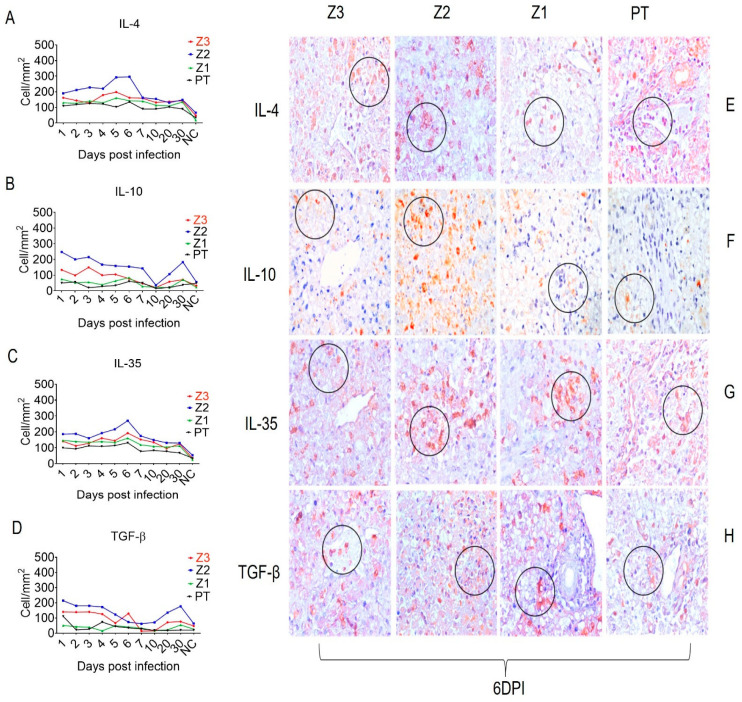
Quantitative immunohistochemical analysis of selected anti- inflammatory cytokines in zones Z3, Z2, Z1 and PT in hepatic parenchyma of squirrel monkeys (*Saimiri* spp.) infected with YFV. Circles represent areas expressing higher immunostaining cells in different hepatic areas. (**A**) Expression of IL-4. (**E**) Immunostaining for IL-4 in hepatocytes (E-Z3), (E-Z2), (E-Z1) in inflammatory infiltrate area (E-PT). (**B**) Expression of IL-10. (**F**) Immunostaining for IL-10 in hepatocytes (F-Z3), (F-Z2), (F-Z1), (F-PT). (**C**) Expression of IL-35. (**G**) Immunostaining for IL-35 in hepatocytes in all examined hepatic tissues (G-Z3), (G-Z2), (G-Z1), (G-PT). (**D**) Expression of TGF-β. (**H**) Immunostaining for TGF-β in the inflammatory infiltrate (H-Z3), (H-Z2), and in hepatocytes (H-Z1), (H-PT) 6 dpi.

**Figure 7 viruses-15-00551-f007:**
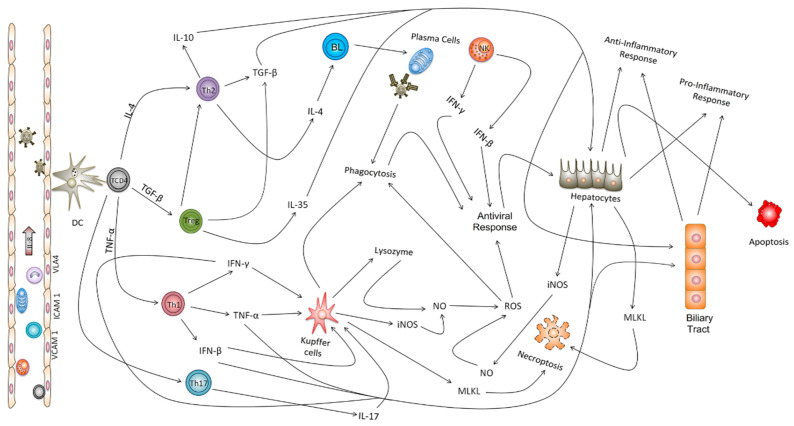
Integrated network and possible mechanism of in situ immune response in the hepatic parenchyma of non-human primates (*Saimiri* spp.) infected with the yellow fever virus. After the infection, YFV is recognized by dendritic cells (DC), initiating the immune response by triggering immune cells that will produce the mechanisms to activate different local cells in liver and the production of cytokines and chemokines to drive the organism defenses. Both innate and adaptive immune cells have an important role in triggering the adequate response and the Treg cells are directly involved in the regulation of an equilibrated immunologic response in efforts to control the damage caused by YFV in the hepatic tissue.

## Data Availability

Data can be requested from the corresponding author.

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
