# Peer review of "Experimental Yellow Fever in Squirrel Monkey: Characterization of Liver In Situ Immune Response"

_viruses, 2023, doi:10.3390/v15020551_

Round 1

Reviewer 1 Report

The study focuses on the characterization of the liver in-situ immune response against YFV in an experimental model of Squirrel Monkey. YF has caused recent outbreaks of great importance, mainly in Latin American countries, with risks of re-urbanization, which demonstrates the importance of this type of study.

The authors used some acronyms in the text without a meaning.

Line 70: “CENP”

Line 77: “IEC”

Line 74: “0.5 ml infective dose containing, 5.31 × 105 copies/ml” How to explain the viral load used?

Line 76: “was originally obtained from a severe fatal human case (BeH655417)” When did the case occur? Where? Has the viral strain been sequenced? The sample was used in an experimental model and needs to be better characterized.

Lines 97-98: “Anti-YFV antibody was produced by the IEC and included a streptavidin–alkaline phosphatase assay.” Is it possible to know for which viral antigens?

Lines 113: “2.4. Semi-quantitative analysis” Here in the Materials and Methods section, the analysis is treated as semi-quantitative, and the Results as quantitative. It is necessary to standardize.

Lines 118-119 “We calculated the mean number of cells in each area and divided the result by 0.0625 mm2” The standard deviation of this analysis should appear on the graphs throughout the study. Make this correction or justify the lack of a standard deviation in the study's essential and central data. I did not have access to Supplementary Materials, and it would be necessary to evaluate the negative control.

Results

The figures are central to this work and must be of higher quality.

Figure 1 - What is NC? It is not mentioned in the caption. Need to complete. E) There is no arrow as indicated in the legend. F) The figure has an arrow that is not indicated in the legend. By the way, arrows are hard to see.

Line 171: Replace “portal tract” to “portal tract (PT)”

Line 177: “S100, CD11b, CD57, CD4, and CD20” Mention, at some point, the reason for choosing the analyzed markers. This data only appears in the Discussion.

Figure 2 - In the graphics legend (A-E), EP appears in the black line. Would it be PT? (F-I) Black circles are not pointed in the legend.

Figure 3 – (C and F) I didn't understand the correlation between the results. Immunostaining in F demonstrates intense coloration, but the graph in C does not support this fact. Could it be a background effect? The authors can add the point in the Discussion.

Figure 4 - I don't know if the drawn circles are the best representation in these figures. In some cases, the detection appears more punctual, but in others, it is more widespread in the tissue presented. In some, the circles should be more prominent; in others, they should be smaller; in others, they should not exist (replace with an arrow, for example). It's a suggestion.

Figure 5 - Subtitles again with EP instead of PT.

Line 366: “protective anti-YFV neutralizing antibodies” Did the authors consider doing PRNT for this study? It would be exciting data.

Figure 7 - It was not pointed out in the text and is in Portuguese.

Author Response

The study focuses on the characterization of the liver in-situ immune response against YFV in an experimental model of Squirrel Monkey. YF has caused recent outbreaks of great importance, mainly in Latin American countries, with risks of re-urbanization, which demonstrates the importance of this type of study.

R: Thank you for your comments.

The authors used some acronyms in the text without a meaning.

Line 70: “CENP”

R: request answered, see text

Line 77: “IEC”

R: request answered, see text

Line 74: “0.5 ml infective dose containing, 5.31 × 105 copies/ml” How to explain the viral load used?

R: A dose infectant of 0.5 ml infective dose containing, 5.31 × 105 copies/ml” [infectious dose: 1 × 106 plaque forming units (PFU)/mL - we modify the text!) was used according to the reference of Monath and Barrett (2003) that demonstrated that the amount of YFV inoculated during the vector bite is 103 to 104 PFU/mL

MONATH TP, BARRET AD. Pathogenesis and pathophysiology of yellow fever. Advances in Virus Research 60: 343–395, 2003.

Line 76: “was originally obtained from a severe fatal human case (BeH655417)” When did the case occur? Where? Has the viral strain been sequenced? The sample was used in an experimental model and needs to be better characterized.

R: request answered, see text.

The YFV sample used in this study was originally obtained from a severe fatal human case in Roraima, Brazil in 2014 of South American genotype I (BeH655417) and isolated in the Evandro Chagas Institute (IEC), Ananindeua, Pará, Brazil and propagated by a single additional passage in C6/36 cells.

Lines 97-98: “Anti-YFV antibody was produced by the IEC and included a streptavidin–alkaline phosphatase assay.” Is it possible to know for which viral antigens?

R: Hyperimmune polyclonal mouse antibody was produced in the Evandro Chagas Institute from a purified viral suspension containing mains flavivirus and CHIKV circulating in the country and that was inoculated in BALBc mice to obtain hyperimmune mouse ascitic fluid for arbovirus differentiation. Hyperimmune polyclonal mouse antibody used recognizes portions of the capsid and envelope proteins of the virus. We are informed that several dilutions were performed and in the minors possible the results were positive for fatal cases. Processing followed the protocol adapted from similar methodologies that used these techniques please see: PMID: 27372395; PMID: 30487475; PMCID: PMC7651848. The negative control used in this study did not show positivity for any of the viruses tested.

Lines 113: “2.4. Semi-quantitative analysis” Here in the Materials and Methods section, the analysis is treated as semi-quantitative, and the Results as quantitative. It is necessary to standardize.

R: Text corrected lines 115-116

Lines 118-119 “We calculated the mean number of cells in each area and divided the result by 0.0625 mm2” The standard deviation of this analysis should appear on the graphs throughout the study. Make this correction or justify the lack of a standard deviation in the study's essential and central data. I did not have access to Supplementary Materials, and it would be necessary to evaluate the negative control.

R: We chose to include the data as supplementary material.

Results

The figures are central to this work and must be of higher quality.

R: The images are found in high resolution in accordance with the standards of the Journal and we also emphasize that for each of the figures, we include areas that are delimited, not circled, signaling morphological aspects of non-woven alteration or a temporary image representation to facilitate the reader's understanding. .

Figure 1 - What is NC? It is not mentioned in the caption. Need to complete. E) There is no arrow as indicated in the legend. F) The figure has an arrow that is not indicated in the legend. By the way, arrows are hard to see.

R: The text and figure have been corrected

R: Text corrected lines 175-176

Line 171: Replace “portal tract” to “portal tract (PT)”

R: Text corrected lines 173-174

Line 177: “S100, CD11b, CD57, CD4, and CD20” Mention, at some point, the reason for choosing the analyzed markers. This data only appears in the Discussion.

R: Thank very much for comments. We have included this information in the manuscript. Line 180 – 187.

R: Thank very much for comments. We include this information in the manuscript

Correctly, the immunomarking for the 4 compartments analyzed maintains a linearity, reinforcing that there was a predominance in zone 2 as observed in the figure where the marking is punctual. The delimitation of the circle in the image illustrates this question.

Figure 2 - In the graphics legend (A-E), EP appears in the black line. Would it be PT? (F-I) Black circles are not pointed in the legend.

R:Figures 2, 5 and 6 were corrected;

Figure 3 – (C and F) I didn't understand the correlation between the results. Immunostaining in F demonstrates intense coloration, but the graph in C does not support this fact. Could it be a background effect? The authors can add the point in the Discussion.

Correctly, the immunomarking for the 4 compartments analyzed maintains a linearity, reinforcing that there was a predominance in zone 2 as observed in the figure where the marking is punctual. The delimitation of the circle in the image illustrates this question.

Figure 4 - I don't know if the drawn circles are the best representation in these figures. In some cases, the detection appears more punctual, but in others, it is more widespread in the tissue presented. In some, the circles should be more prominent; in others, they should be smaller; in others, they should not exist (replace with an arrow, for example). It's a suggestion.

R: We appreciate your comments. However, we believe that the circle delimits more of the visual field for the reader around the histopathological alterations that are observed and the present immunomarking, and helps readers especially non-specialist in pathology.

Figure 5 - Subtitles again with EP instead of PT.

R:Figures 2, 5 and 6 were corrected;

Line 366: “protective anti-YFV neutralizing antibodies” Did the authors consider doing PRNT for this study? It would be exciting data.

R: We were unable to standardize or PRNT test for Saimiri sp. The results obtained do not corroborate with the results of the other methodologies used. We performed neutralization tests worldwide and the data is presented in another study submitted to Viruses - not published yet.

Figure 7 - It was not pointed out in the text and is in Portuguese.

R: As corrections were made as needed.

Reviewer 2 Report

The article by Ferreira and colleagues, “Experimental Yellow Fever in Squirrel Monkey:  Characterization of liver in-situ immune response”, details the authors’ investigation into the liver pathology of the yellow fever virus (YFV) infected squirrel monkey.   For their study, the authors perform immunostaining of the livers of the infected non-human primates (NHP) over a time course from 1-30 days post infection.  The overall conclusion, which is generally supported by the data, is that the YFV-infected squirrel monkey exhibits a hepatic pathogenesis similar to that seen in humans and thus may be a good model for human YFV infection.

The paper is generally well-written and the data support the conclusions presented.  The primary unaddressed caveat to this manuscript is that the presented figures are apparently all derived from the single NHP that succumbed to the disease rather than being euthanized at a given time point.  One can then infer that NHP had more severe disease and likely presented with a response that was different, (i.e. more or less intense) than the other NHPs.  Indeed, this is borne out when looking at the time course graphs in Figs 2-6.  In many cases, 6 dpi is clearly an outlier relative to the rest of the time course. For extreme examples, see Fig 2E and Fig 4B.  Possibly the authors deliberately chose this more severe case for detailed analysis, but at a minimum, this should be mentioned and addressed in the text.

Finally, while having negative controls to compare side by side would have been useful, the authors do mention that the negative control is included in the supplementary data.  However, there was no available link to access supplementary data, thus this could not be evaluated.

Other comments by line number:

18:  “Yellow Fever Virus” should not be capitalized.

20:  “Saimiri spp.” should be italicized.

38:  Errant “)” after “Organization”.

51:  Suggest noting that these two citations are essentially self-citations since there are several common authors between these papers and the current manuscript.

58:  Citations should use brackets instead of parentheses.

64 (and throughout):  “in-situ” should not be hyphenated and should probably be in italics (unless MDPI journals specify otherwise)

70:  Write out CENP.

72:  Supplementary table not available for review.  Also not mentioned in supplementary materials (line 395) at the end of the manuscript.

73:  Possibly the supplementary table would have shown, but how many animals showed no immunity to YFV and cross-reacting flaviviruses?  There are ten time points and a negative control in the data, so is it assumed that none of the eleven had any cross-reactive antibodies?

74:  Was the control mock inoculated?  If so, on what day post mock inoculation was the control euthanized?

77:  Write out IEC.

155 (Figure 1):  It is assumed that NC is negative control, but suggest explicitly saying that in the figure legend.

171:  Suggest changing ‘red circles’ to ‘red circle’.

172:  Suggest changing ‘circle’ to ‘clack circle’ for consistency.

184:  Line 179 notes that CD20 was absent on d6 and that is corroborated but figure 2E and figure 2J which does not show any staining in any if the four panels.  Therefore, 2J does not support the statement in line 184 that CD20 was more highly expressed in Z2.

185 (figure 2):  “TP” should be “PT” at the top of the right-most set of panels.

187:  “Squirrel” should not be capitalized here or in the legends for all subsequent figures.

227 (figure 5):  What is EP at the top of the right-most set of panels?  Same question for figure 6.

252-253:  Figure 1B says that viral RNA was detected at a level of 10^12 copies/uL at 30 dpi.  How does that fit with the statement that from day 7 onward viral RNA was not detected?

383 (figure 7):  Please prepare the figure in English.

Author Response

The article by Ferreira and colleagues, “Experimental Yellow Fever in Squirrel Monkey:Characterization of liver in-situ immune response”, details the authors’ investigation into the liver pathology of the yellow fever virus (YFV) infected squirrel monkey.   For their study, the authors perform immunostaining of the livers of the infected non-human primates (NHP) over a time course from 1-30 days post infection.  The overall conclusion, which is generally supported by the data, is that the YFV-infected squirrel monkey exhibits a hepatic pathogenesis similar to that seen in humans and thus may be a good model for human YFV infection.

The paper is generally well-written and the data support the conclusions presented.  The primary unaddressed caveat to this manuscript is that the presented figures are apparently all derived from the single NHP that succumbed to the disease rather than being euthanized at a given time point.  One can then infer that NHP had more severe disease and likely presented with a response that was different, (i.e. more or less intense) than the other NHPs.  Indeed, this is borne out when looking at the time course graphs in Figs 2-6.  In many cases, 6 dpi is clearly an outlier relative to the rest of the time course. For extreme examples, see Fig 2E and Fig 4B.  Possibly the authors deliberately chose this more severe case for detailed analysis, but at a minimum, this should be mentioned and addressed in the text.

Finally, while having negative controls to compare side by side would have been useful, the authors do mention that the negative control is included in the supplementary data.  However, there was no available link to access supplementary data, thus this could not be evaluated.

R: Thank you for your comments. We have address accordingly. Please, see the new version of the manuscript.

Other comments by line number:

18: “Yellow Fever Virus” should not be capitalized.

R: request answered, see text.

20: “Saimiri spp.” should be italicized.

R: request answered, see text.

38:  Errant “)” after “Organization”.

R: request answered, see text.

51:  Suggest noting that these two citations are essentially self-citations since there are several common authors between these papers and the current manuscript.

R: We appreciate or comment, but we credit that the references are relevant for the study

58:  Citations should use brackets instead of parentheses.

R: request answered, see text.

64 (and throughout): “in-situ” should not be hyphenated and should probably be in italics (unless MDPI journals specify otherwise)

R: request answered, see text.

70:  Write out CENP.

R: request answered, see text.

72:Supplementary table not available for review.Also not mentioned in supplementary materials (line 395) at the end of the manuscript.

R: As corrections were made according to suggestion.

73:  Possibly the supplementary table would have shown, but how many animals showed no immunity to YFV and cross-reacting flaviviruses?  There are ten time points and a negative control in the data, so is it assumed that none of the eleven had any cross-reactive antibodies?

R: request answered, see text.

Before the infection all animals were bled and tested negative by hemagglutination-inhibition (HI) test for the presence of antibodies against YFV, dengue virus 1-4, Ilheus virus, Rocio virus, Sant Louis virus.  (Lines 71-73). After infection, the animals were tested again for HI and as soon as those infected experimentally presented antibodies for YFV and cross-reaction for dengue virus 1, 2, 3, 4; Ilheus virus; Saint Louis Virus Encephalitis; Rocio virus; Data submitted to Viruses - another paper.

74:  Was the control mock inoculated?  If so, on what day post mock inoculation was the control euthanized?

R: The control was not inoculated! It was sacrificed at the end of the experimental kinetics (31 days after the onset of the experiment)

77:  Write out IEC.

R: request answered, see text.

155 (Figure 1):  It is assumed that NC is negative control but suggest explicitly saying that in the figure legend.

R: request answered, see text.

171:  Suggest changing ‘red circles’ to ‘red circle’.

R: request answered, see text.

172:  Suggest changing ‘circle’ to ‘clack circle’ for consistency.

R: request answered, see text.

184:Line 179 notes that CD20 was absent on d6 and that is corroborated but figure 2E and figure 2J which does not show any staining in any if the four panels.  Therefore, 2J does not support the statement in line 184 that CD20 was more highly expressed in Z2.

R: As corrections were made as requested.

185 (figure 2): “TP” should be “PT” at the top of the right-most set of panels.

R:Figures 2, 5 and 6 were corrected;

187:“Squirrel” should not be capitalized here or in the legends for all subsequent figures.

R: request answered, see text.

227 (figure 5):  What is EP at the top of the right-most set of panels?  Same question for figure 6.

R:Figures 2, 5 and 6 were corrected;

252-253:  Figure 1B says that viral RNA was detected at a level of 10^12 copies/uL at 30 dpi.  How does that fit with the statement that from day 7 onward viral RNA was not detected?

R: Text corrected lines 257-258

383 (figure 7):  Please prepare the figure in English.

R: As corrections were made as needed.

Reviewer 3 Report

The submitted manuscript by Ferreira et al describes analysis of liver tissues collected from a small number of squirrel monkeys experimentally infected with yellow fever virus (YFV) in a natural history study.  Liver sections were examined histologically and were evaluated by immunohistochemistry for a range of immunological and cellular markers.  Data was analyzed based on individual zones within the liver and visual data from an animal that died from infection at day 6 post-infection is provided as exemplar data.  Overall, these data showed morphologic liver characteristics similar to that described in human disease.  There was evidence of a range of immunologic markers in tissues collected from n=1 animal at each of the sample collection points with some differences based on the zone analyzed.

While I think the data provided here is useful and adds to our understanding of yellow fever pathogenesis, the limited number of samples makes it difficult to interpret the data in the context of a heterogeneous population.  While I am not suggesting additional experimental work with YFV infected animals, I think it is important to include data from an uninfected control animal and to modify the discussion and conclusion sections to clearly recognize the limitations of the small sample set.

It is also important to justify the use of this particular species for these studies including information on the natural disease course following YFV infection.  In the studies described here, only one animal succumbed ‘naturally’ from the cohort of 11.  Is this expected?  What is the typical disease course/outcome in YFV infected squirrel monkeys and how is this relevant to human disease?  Also, animal care information including any accreditation and animal use protocol information should be provided.

Specific comments:

Overall, this manuscript is well written.  However, there are still some typos so a careful review is necessary.

Lines 56-65:  Add rationale for the use of squirrel monkeys

Lines 72-73:  Can the Ferreira et al ‘in press’ citation be updated?

Lines 90-91: “…it was used…”is improper language use.  Please modify.

Lines 95-111”  Please provide a list or table of the antigenic markers targeted, the antibodies (and clones) used and the source from which they were acquired.

Lines 129-130:  What is the natural disease course in YFV infected squirrel monkeys?

Line 154:  I am not sure what is meant by “…regressed up to…”

Figure 1:  The legend indicates there should be an arrow, but I don’t see one.

Figure 2 in particular, and others. The authors should identify better color schemes for arrows and circles in the images.  In some cases, the markings are barely visible.

Lines 253-259:  What about YFV in squirrel monkeys?

Lines 260-262:  The preferential cell populations here seem to cover most cells in the liver.  Can these be ranked?  For example, “YFV appears to primarily infect hepatocytes in the liver of squirrel monkeys, although Kupffer cells are also a major target.”

Line 381:  Apoptosis is a component of ‘programmed cell death’, but virus induced apoptosis is just that.  It is not programmed cell death.

Figure 7 is in Portuguese.  It should be modified to English.

Author Response

The submitted manuscript by Ferreira et al describes analysis of liver tissues collected from a small number of squirrel monkeys experimentally infected with yellow fever virus (YFV) in a natural history study.  Liver sections were examined histologically and were evaluated by immunohistochemistry for a range of immunological and cellular markers.  Data was analyzed based on individual zones within the liver and visual data from an animal that died from infection at day 6 post-infection is provided as exemplar data.  Overall, these data showed morphologic liver characteristics similar to that described in human disease.  There was evidence of a range of immunologic markers in tissues collected from n=1 animal at each of the sample collection points with some differences based on the zone analyzed.

While I think the data provided here is useful and adds to our understanding of yellow fever pathogenesis, the limited number of samples makes it difficult to interpret the data in the context of a heterogeneous population.  While I am not suggesting additional experimental work with YFV infected animals, I think it is important to include data from an uninfected control animal and to modify the discussion and conclusion sections to clearly recognize the limitations of the small sample set.

It is also important to justify the use of this particular species for these studies including information on the natural disease course following YFV infection.  In the studies described here, only one animal succumbed ‘naturally’ from the cohort of 11.  Is this expected?  What is the typical disease course/outcome in YFV infected squirrel monkeys and how is this relevant to human disease?  Also, animal care information including any accreditation and animal use protocol information should be provided.

R: Thank you very much for your comments. The manuscript was edited and changed as needed.

Specific comments:

Overall, this manuscript is well written.  However, there are still some typos so a careful review is necessary.

Lines 56-65:  Add rationale for the use of squirrel monkeys

R:Request answered, see text.

Lines 72-73:  Can the Ferreira et al ‘in press’ citation be updated?

R:We choose to delete the information, since the manuscript is not yet published.

Lines 90-91: “…it was used…”is improper language use.  Please modify.

R:Request answered, see text.

Lines 95-111:” Please provide a list or table of the antigenic markers targeted, the antibodies (and clones) used and the source from which they were acquired.

R:Request answered, see text.

Lines 129-130:  What is the natural disease course in YFV infected squirrel monkeys?

R: Squirrel monkeys (Saimiri genus) have moderate sensitivity to YFV when compared to other monkeys genera such as the Alouatta genus. In general the majority of squirrel monkeys survive the YFV infection, as presented in this study.

Line 154:  I am not sure what is meant by “…regressed up to…”

R:Request answered, see text.

Figure 1:The legend indicates there should be an arrow, but I don’t see one.

R: Request answered, see text.

Figure 2, and others. The authors should identify better color schemes for arrows and circles in the images.  In some cases, the markings are barely visible.

R: Thank you for your comment. The Images are in high resolution according to the journal's rules and we also emphasize that for each of the figures, we include areas that are delimited in the circle signaling the representation of very specific immunostaining to facilitate the reader's understanding, and the readers especially the non-specialists in pathology.

Lines 253-259:What about YFV in squirrel monkeys?

R:Request answered, see text.

Lines 260-262:  The preferential cell populations here seem to cover most cells in the liver.  Can these be ranked?  For example, “YFV appears to primarily infect hepatocytes in the liver of squirrel monkeys, although Kupffer cells are also a major target.”

R: In fact, it is the resident cells that make up a major group of cells for which the virus has a tropism, which we have highlighted in our manuscript.

Line 381: Apoptosis is a component of ‘programmed cell death’, but virus induced apoptosis is just that.  It is not programmed cell death.

R: Thank you for your comment. We modified the sentence to improve understanding.

Figure 7 is in Portuguese.It should be modified to English.

R: Figure 7 now is in English as needed.

Reviewer 4 Report

In the manuscript "Experimental Yellow Fever in Squirrel Monkey: Characterization of liver in-situ immune response" the authors describe pathological and immunological characteristics of yellow fever infection in a non-human primates (Saimiri spp.) model. They showed that this NHP is susceptible to YF infection and develops liver lesions and immune responses that can be associated with YF pathogenesis' in humans. The study was well conducted but I have a few points to clarify, which is why I am recommending major revisions.

ABSTRACT: is it necessary to divide the abstract into Method, Results and Conclusion topics? If it is not mandatory, I would recommend the authors remove these sub-titles. Also, italicized the species names.

INTRODUCTION

Line 42:  What is "high mortality rate"? Please, describe a range number to this.

Line 56: the authors cite the risk of reurbanization of YFV. I missed a sentence explaining the two different cycles that exist in Brazil, which would help to explain the risk of reurbanization.

Line 58-63: why the study of YF pathogenesis would help in the understanding of a "serious viscerotropic disease following vaccination"? This study can help patient management in general. I can't see why this would be different. The authors are not using YFV-17DD; which would explain that the analysis could only be extrapolated to a vaccine adverse effects.

M&M

Line 70: What is CENP? Describe the acronym.

Line 72: because the manuscript (Ferreira et al)  is not published yet, I recommend the authors briefly describe it here. 

Line 74: Is this "BeH655417" the GenBank accession number? Please, clarify it in the manuscript text.

Line 74: 5.31x10E6 copies of what? Genome? It needs to be clarified. If it is genome copies, Do you have the correlation in PFU? This data is more interesting since the genome copies do not indicate how many available particles will infect the animals.

Line 77: what is IEC? 

Line 78-79: which infection was confirmed by this method? During the virus isolation at IEC or after animal inoculation? This part is a little bit confusing and needs to be clarified in the manuscript.

Line 90-93: Describe the quantification protocol a bit more: what part of the YFV genome was targeted? Did you do a previous PCR only for YFV detection or the only PCR done was this one that also quantifies copies of the genome?

Sub-topic 2.4: semi-quantitative analysis of what? Describe it better.

RESULTS:

The quality of images needs to be improved.

Add the standard bars to each graph.

Each result per day here represents an N=1, right? This point needs to be better clarified in the manuscript.

Line 129-131: When does the convalescent phase start? After reading a few papers on YF, it seems that it starts after 15 days post symptoms (dps). In the manuscript you're saying in days post infection - is there any correlation with dps? Also, the authors mentioned 2-10 dpi and described it as a convalescent phase - isn't it too early to be a convalescent phasE? I know the days of each phase of a YF infection can be a bit confusing, but I want to know how authors are classifying/dividing the disease in this manuscript.

Figure 1: The authors presented here figures from the only animal who died due to YFV after the infection. I'm wondering if there is any difference among this animal and the others which were euthanized.

Figure 2: letters in the text and in the figure are different - please, review them.

DISCUSSION

The discussion was well elaborate and I don't have any point to add.

Figure 7: Please, translate Portuguese names to English.

Author Response

In the manuscript "Experimental Yellow Fever in Squirrel Monkey: Characterization of liver in-situ immune response" the authors describe pathological and immunological characteristics of yellow fever infection in a non-human primates (Saimiri spp.) model. They showed that this NHP is susceptible to YF infection and develops liver lesions and immune responses that can be associated with YF pathogenesis' in humans. The study was well conducted but I have a few points to clarify, which is why I am recommending major revisions.

 R: Thank you for your comments.

ABSTRACT: is it necessary to divide the abstract into Method, Results and Conclusion topics? If it is not mandatory, I would recommend the authors remove these sub-titles. Also, italicized the species names.

R:Request answered, see text.

INTRODUCTION

Line 42:  What is "high mortality rate"? Please, describe a range number to this.

R:Request answered, see text.

Line 56: the authors cite the risk of reurbanization of YFV. I missed a sentence explaining the two different cycles that exist in Brazil, which would help to explain the risk of reurbanization.

R:Request answered, see text.

Line 58-63: why the study of YF pathogenesis would help in the understanding of a "serious viscerotropic disease following vaccination"? This study can help patient management in general. I can't see why this would be different. The authors are not using YFV-17DD; which would explain that the analysis could only be extrapolated to a vaccine adverse effect.

R: Thank you for your comment. Request answered, see text.

M&M

Line 70: What is CENP? Describe the acronym.

R:Request answered, see text.

Line 72: because the manuscript (Ferreira et al)is not published yet, I recommend the authors briefly describe it here. 

R:We chose to delete the citation

Line 74: Is this "BeH655417" the GenBank accession number? Please, clarify it in the manuscript text.

R:This is the identification of the institutional sample - IEC

Line 74: 5.31x10E6 copies of what? Genome? It needs to be clarified. If it is genome copies, Do you have the correlation in PFU? This data is more interesting since the genome copies do not indicate how many available particles will infect the animals.

R:Request answered, see text.

Line 77: what is IEC? 

R:Request answered, see text.

Line 78-79: which infection was confirmed by this method? During the virus isolation at IEC or after animal inoculation? This part is a little bit confusing and needs to be clarified in the manuscript.

R: Cellular infection by YFV was confirmed by IFI before experimental infection of the animals.

Line 90-93: Describe the quantification protocol a bit more: what part of the YFV genome was targeted? Did you do a previous PCR only for YFV detection or the only PCR done was this one that also quantifies copies of the genome?

For the detection and quantification of viral RNA, a single-step RT-qPCR was conducted with the probe and primers described by Domingo et al. (2012) (Table XX), which bind in the 5'-UTR region, at 200 and 300 nM, respectively. It was carried out using the ViiA 7 Real-Time PCR System equipment (Life Technologies/USA) and the Taqman One Step PCR Master Mix kit (Life Technologies/USA), per the manufacturer's instructions. Using controls containing previously cloned plasmids containing the 5'-UTR region of the YFV, an absolute quantification was done.

Table XX - Primers and probe used for YFV quantification and detection

Primer orprobe

Sequence (5’-3’)

Position

YFallF

GCTAATTGAGGTGYATTGGTCTGC

15–38

YFallR

CTGCTAATCGCTCAAMGAACG

83–103

YFallP

FAM-ATCGAGTTGCTAGGCAATAAA CAC-BHQ2-3

41–64

Sub-topic 2.4: semi-quantitative analysis of what? Describe it better.

R:corrected text

RESULTS:

The quality of images needs to be improved.

R: We disagree. The images are in high resolution according to the Viruses standards and we also emphasize that for each of the figures, we include areas that are delimited in the circle signaling morphological aspects of tissue alteration or the representation of very specific immunostaining to facilitate the reader's understanding

Add the standard bars to each graph.

R: Request answered, see figure

Each result per day here represents an N=1, right? This point needs to be better clarified in the manuscript.

R: Request answered, see text.

Line 129-131: When does the convalescent phase start? After reading a few papers on YF, it seems that it starts after 15 days post symptoms (dps). In the manuscript you're saying in days post infection - is there any correlation with dps? Also, the authors mentioned 2-10 dpi and described it as a convalescent phase - isn't it too early to be a convalescent phasE? I know the days of each phase of a YF infection can be a bit confusing, but I want to know how authors are classifying/dividing the disease in this manuscript.

R:Thanks for the comment. The suggestion has been corrected in the text.

Figure 1: The authors presented here figures from the only animal who died due to YFV after the infection. I'm wondering if there is any difference among this animal and the others which were euthanized.

R: Ferreira MS, Júnior PSB, Cerqueira VD, Rivero GRC, Júnior CAO, Castro PHG, Silva GAD, Silva WBD, Imbeloni AA, Sousa JR, Araújo APS, Silva FAE, Tesh RB, Quaresma JAS, Vasconcelos PFDC. Experimental yellow fever virus infection in the squirrel monkey (Saimiri spp.) I: gross anatomical and histopathological findings in organs at necropsy. Mem Inst Oswaldo Cruz. 2020 Nov 9;115:e190501. doi: 10.1590/0074-02760190501. PMID: 33174908; PMCID: PMC7651848.

Figure 2: letters in the text and in the figure are different - please, review them.(Jorge)

R: Request answered, see text.

DISCUSSION

The discussion was well elaborate, and I don't have any point to add.

R: Thank you for your comment, we appreciate it.

Figure 7: Please, translate Portuguese names to English.

R: Figure 7 is now in English as needed.

Round 2

Reviewer 1 Report

The authors have made considerable improvements. I have no other suggestions/recommendations to make.

Author Response

Thanks! We attach the file with the latest revision!

Reviewer 4 Report

The authors made a substantial improvement to the text. 

However, I still have minor revisions that should be addressed, listed below:

Line 68: replace VFA with YFV

Figure 1 A, B, and C are blurred and need improvement. I still think that all graph's figures need some improvement. They are all blurred a little bit. 

Looking at Figure 1B and comparing it to the legend, I do not think the authors can say the peak was at 4dpi - the values at 7 dpi are similar. This point needs to be clarified in the manuscript - Why are the authors considering 4dpi as the peak?

Line 168: Remove the word "and" from this sentence "wide predominance in Z2 and during the acute phase" 

Author Response

Thanks for your comments. Adjustments were made as requested.

The authors made a substantial improvement to the text. 

However, I still have minor revisions that should be addressed, listed below:

Line 68: replace VFA with YFV

R: Thanks. The text has been corrected!

Figure 1 A, B, and C are blurred and need improvement. I still think that all graph's figures need some improvement. They are all blurred a little bit. 

R: We made the corrections as previously requested. Figures are as recommended by the journal.

Looking at Figure 1B and comparing it to the legend, I do not think the authors can say the peak was at 4dpi - the values at 7 dpi are similar. This point needs to be clarified in the manuscript - Why are the authors considering 4dpi as the peak?

R: Thanks for the considerations! The figure has been corrected!

Line 168: Remove the word "and" from this sentence "wide predominance in Z2 and during the acute phase" 

R: Thanks. The text has been corrected!
